# New Trends in Diaziridine Formation and Transformation (a Review)

**DOI:** 10.3390/molecules26154496

**Published:** 2021-07-26

**Authors:** Zetryana Puteri Tachrim, Lei Wang, Yuta Murai, Makoto Hashimoto

**Affiliations:** 1Division of Applied Bioscience, Graduate School of Agriculture, Hokkaido University, Kita 9, Nishi 9, Kita-ku, Sapporo 060-8589, Japan; leiwang@dlut.edu.cn (L.W.); ymurai@sci.hokudai.ac.jp (Y.M.); 2Research Center for Chemistry, Indonesian Institute of Sciences, Kawasan Puspiptek, Serpong, South Tangerang 15314, Banten, Indonesia; 3State Key Laboratory of Fine Chemicals, Department of Pharmacy, School of Chemical Engineering, Dalian University of Technology, Dalian 116024, China; 4Frontier Research Center for Post-Genome Science and Technology, Faculty of Advanced Life Science, Hokkaido University, Kita 21, Nishi 11, Kita-ku, Sapporo 001-0021, Japan

**Keywords:** diaziridines, substituted diaziridines, diazirines, formation, synthesis, transformation, reactivity

## Abstract

This review focuses on diaziridine, a high strained three-membered heterocycle with two nitrogen atoms that plays an important role as one of the most important precursors of diazirine photoaffinity probes, as well as their formation and transformation. Recent research trends can be grouped into three categories, based on whether they have examined non-substituted, *N*-monosubstituted, or *N*,*N*-disubstituted diaziridines. The discussion expands on the conventional methods for recent applications, the current spread of studies, and the unconventional synthesis approaches arising over the last decade of publications.

## 1. Introduction

An important intermediate and precursor in organic chemistry, diaziridine [1] is a highly strained three-membered heterocycle with two nitrogen atoms [2]. The other excellent properties of diaziridine include a weak N–N bond, low toxicity, hydrazine aminal duality [3], and neurotropic activity [4]. Diaziridines, independently discovered by three research groups (Schmitz’s, Paulsen’s, and Abendroth and Henrich’s research groups) between 1958 and 1959, progressively achieved popularity in the field of chemistry over the two decades following these first publications [5]. Non-substituted diaziridine (diaziridine without any substituent on the nitrogen atom, **1**) was developed as a reagent for a variety of oxidative transformations [6] and employed in the synthesis of carbene-mediated photoaffinity-labeled photophores of 3*H*-diazirines [7,8,9,10]. The trend of using diaziridine **1** as a precursor for the diazirine photoaffinity probe has been progressively updated until the present day, and a series of reviews have discussed it since the 2010s [7,11,12,13,14,15]. Although the chemistry of *N*-monosubstituted diaziridine **2** has not been studied as extensively as that of non-substituted diaziridine **1**, there are examples documenting its application as an *N*-transfer reagent with α,β-unsaturated amides to form stereopure aziridines [16]. As for *N*,*N*-disubstituted diaziridine **3**, these valuable heterocycles have also been used as a versatile reagent via selective C–N or N–N cleavage in organic synthesis, particularly their bicyclic analogues of 1,5-diazabicyclo[3.1.0]-hexanes, which possess a strained cis-*N*,*N*-disubstituted diaziridine fragment for 1,3-dipolar cycloaddition [17]. The advancements in substituted diaziridine with respect to its nitrogen atom(s) synthesis and applications have been summarized in several reviews [3,4,5,18,19,20]. On the basis of these excellent reviews, only a few studies have been fully dedicated to investigating the potential properties diaziridine with respect to its monocyclic synthetic methods [20] or its reactivity towards electrophilic reagents [5]. Here in this review, the scope is limited to the current formation and transformation of diaziridines. Then, the trends in these areas are simplified into three categories—non-, *N*-mono-, and *N*,*N*-disubstituted diaziridines (Figure 1)—in which the discussion describes recent conventional uses, the current field of studies, and unconventional approaches that have emerged over the last decade of publications.

## 2. The Chemistry of Diaziridines

Three key approaches are known for the construction of monocyclic diaziridines **4**, and these were summarized by Makhova et al. in their previous 2008 review of diaziridine (Figure 2) [20]. First, the reaction of primary aliphatic amines or ammonia **5** with the condensation products of carbonyl compounds **6** and aminating reagents. Second, the reaction of imines **7** (the condensation product of carbonyl compounds and primary aliphatic amines) is conducted with aminating reagent **8**. Last, a three-component condensation reaction involving the carbonyl compound **9**, primary aliphatic amines or ammonia **10**, and aminating reagents (hydroxylamine-O-sulfonic acid (HOSA) or halo(alkyl)amine, **11**). In addition, in their summary, Meijler et al. [7] emphasized that diaziridine synthesis as a precursor of 3-aryl-3-trifluoromethyl-3*H*-diazirine can start from the preparation of corresponding α,α,α-trifluoroacetophenone via oximation and tosylation (or mesylation)—the condensation products of carbonyl compounds **6**—followed by the treatment of ammonia. In consequence, these known approaches can be described as the “conventional” synthesis methods that have generally been used for the synthesis of diaziridine up until the most recent past. The reactivity of diaziridine was then pursued in order to broaden the advanced applications of these three-membered heterocycles, with reactions that can take place with or without any ring cleavage [1], or the transformation of its unprotected form into respective diazirines through oxidation. Diaziridine transformation induced by electrophilic reagents started being studied in the early 1950s and these reactions are primarily associated with the steric strain of the diaziridine ring and the presence of its two nitrogen atoms [5].

The rapid improvement in the quality of analysis instrumentation for experimental structure identification and detailed stereocenter has lead to the intensive study of the detailed structure determination of diaziridine. Experimental data regarding the structures of diaziridines are important for achieving deeper insight into the previously elucidated features of the diaziridine ring structure, in particular, for the comparison of its corresponding N-N or C-N bond length [21]. Recently, experimental data were obtained for the structural determination of *N*,*N*-disubstituted diaziridine [22] and its trisubstituted form, which possesses an additional alkyl group at position 3 [23,24], bis-diaziridine [25], and bicyclic diaziridine [26] derivatives by the method of gas electron diffraction. Most diaziridine derivatives are liquids; thus, the recent experimental determination of its structures on the basis of X-ray diffraction studies [21,27] has only been performed in a few cases. Investigation of conformational and configurational changes in order to understand the effect of the interconversion barrier on the steric hindrance of the diaziridine substituent has also been conducted using dynamic gas chromatography [28,29,30]. The identification of equilibrium (molecular) structures using joint analysis of gas phase electron diffraction, quantum chemistry, and spectroscopic data [25,26] is becoming a trend and has made diaziridine stereochemical study more visible. Table 1 summarizes the details of recent experimental structure identification studies of several diaziridine derivatives.

Since the strained ring compound exhibits high detonation performance when releasing a large amount of strain energy during ring opening, diaziridine properties were taken into consideration in this study. The design and theoretical investigation of a series of trinitromethane derivatives of diaziridines as candidates for high-energy-density materials were conducted [31]. A computational study of several properties of diaziridines was performed by using Density Functional Theory—B3LYP functional and the aug-cc-pVDZ basis set with the Gaussian 09 program. The formation heat, energetic properties, stability, impact sensitivity, and the calculation of vibrational frequency were studied, and it was found that the trinitromethane substituent was beneficial for enhancing the detonation properties. Among the diaziridines tested, **12** (Figure 3A) and 1-(trinitromethyl)diaziridine (**13**, Figure 3B) were confirmed to possess good potential as a high-density materials, possessing good detonation performance and higher stability.

## 3. Diaziridine Formation and Transformation

### 3.1. Non-Substituted Diaziridines

#### 3.1.1. Conventional Method for Recent Uses

The concept of using the conventional method for the advanced synthesis and application of diaziridines that are non-substituted at their *N*-terminal started with the use of a programmable batch of synthesis robots to design and develop a diazirine-based photoreactive compound [32]. This recent study mimicked the manual organic synthesis workflow by first synthesizing the 3*H*-diazirine-based crosslinker that mediates the formation of diaziridine from ketone. The automatization was constructed through the addition of methanol-NH_3_ and NH_2_OSO_3_H to the ketone, filtration and evaporation of the resulting diaziridine, all followed by the formation of 3*H*-diazirine by an I_2_-Et_2_O system at low temperature, ready for the next reaction process. The feasibility of the robotic synthesis workflow is possibly due to the convenient formation of diaziridine as an intermediate for the synthesis of diazirine, which can be formed using commercial starting materials and universal reagents. Therefore, the recent automatization of the conventional method for synthesis of 3*H*-diazirine broadens the potential of the formation of diaziridine as an intermediate for advanced uses.

The reactivity of non-substituted diaziridines towards oxidants for the feasible synthesis of 3*H*-diazirines expands its applicability as an important intermediate. In a study exploring the process of Pd(0)-catalyzed cross-coupling of diazirines with aryl halides assisted by microwave irradiation [33], 3-methyl-3-(*p*-tolyl)diaziridine **14** was employed for the Pd(0)-catalyzed cross-coupling with 4-bromotoluene **15**—resulting in the product 4,4’-(ethene-1,1-diyl)bis(methylbenzene) **16**—under oxidative conditions following the addition of Ag_2_O as an oxidant (Scheme 1). The Pd(0)-catalyzed cross-coupling is the first example of a transition-metal-catalyzed reaction of diazirines supported by combination with the conventional oxidation of diaziridine, and significantly expands the chemistry of three-membered heterocycles.

In continuation of the oxidation of diaziridine followed by oxidation into diazirine, it is possible to construct the ^15^N—^15^N moiety when synthesizing the three-membered heterocycle diazirine (Figure 4). The conventional transformation of ketone **17** into diaziridine **18** can result in the formation of ^15^N_2_-diazirine **19** molecular tags—hyperpolarized heteronuclei—contained in the diazirine moiety, supporting its potential as a molecular tag for NMR and MRI [34,35]. The readily oxidized ^15^N_2_-diaziridine **18** was possible to synthesize using ^15^NH_3_ and ^15^N-labeled hydroxylamine *O*-sulfonic acid (^15^NH_2_OSO_3_H), whereby the common I_2_-Et_3_N system was used for the formation of the final ^15^N_2_-diazirine product. Although the overall yield achieved for the synthesized ^15^N_2_-diazirine molecular tag was less than 50%, this is sufficient for further modification and analysis. Therefore, the development of subsequent utilizations of this isotope-labeled compound is to be expected.

The synthesis of spirocyclic 3*H*-diazirine-containing building blocks has been discussed in the literature to a lesser extent [36]; however, it can potentially be employed to probe underexplored parts of proteome with libraries derived from structurally similar scaffolds of the parent’s analogues [13,14,37]. Currently, spirocyclic 3*H*-diazirines can be found in steroid [38,39,40], proline [41,42], the simplified structure of chamuvarinin [43] analogues, and 1,5-disubsituted tetrazoles containing 3*H*-diazirine [44]. In a process with fewer steps, the crude diaziridine formed using the conventional method can be used directly for the synthesis of functionalized spirocyclic 3*H*-diazirine without using the chromatographic technique [36]. The transformation of the five- to seven-membered heterocycle or bicycle ketone **20** (Figure 5A) into diazirine **21** via NH_3_ and NH_2_OSO_3_H can be used for the formation of diaziridine **22**, when subjected to an I_2_-Et_3_N system after filtration and concentration. Despite the use of the conventional NH_2_OSO_3_H as a reagent for the formation of diaziridine as an intermediate, four-membered heterocyclic 3*H*-diazirine **23** can be achieved by the sequential formation of oxime **24** from ketone **25** and *O*-sulfonylation for the formation of diaziridine **26**, followed by oxidation into diazirine **23** (Figure 5B). This method is commonly used for the synthesis of 3-phenyl-3-(trifluoromethyl)-3*H*-diazirine (TPD) instead of spirocyclic (aliphatic skeleton) 3*H*-diazirine. Both conventional methods can be used to achieve scles of up to 50 g when synthesizing functionalized spirocyclic 3*H*-diazirine mediated by diaziridine formation. By using this one-pot approach, two types of spirocyclic compounds (compound **47g**–**h**, Table 2, Entries 7–8) can also be successfully synthesized, and will be discussed in next section.

An ambient light stabile-favored photoaffinity labeling probe comprising 3-pyridyl- and 3-pyrimidyl-substituted 3-trifluoromethyl-diazirines **27** [45] was synthesized from alcohols **28**, which directly transformed into an important scaffold representative, ketone **29** (Scheme 2). The replacement of the phenyl group with an electron-withdrawing pyridine or pyrimidine ring indicates that there were no obstacles during the sequential process of oxime formation and *O*-sulfonylation for diaziridine **30** formation. The conventional methods offered by the study showed that the 3-pyridyl- and 3-pyrimidyl-substituted 3-trifluoromethyl-diaziridines were stable and could be readily oxidized into diazirine **27** with Ag_2_O-ether system.

The novel asymmetric synthesis of trifluoromethyldiazirine-based lactisole derivatives **31** occurs as a result of the preparation of trifluoroacetyl modified on the aromatic ring of lactisole, which then introduces the diazirinyl three-membered ring on the trifluoroacetyl group [46]. However, to synthesize the starting material, advanced methods are required; since the 2-, 3-, and 4-trifluoroacetyl lactisole derivatives **32** (Scheme 3) need to consider the optical retention of the protected lactate counterpart, the conventional diazirine **31** construction via diaziridine **33** formation can be easily conducted, starting with oximation with hydroxylamine hydrochloride, tosylation for the hydroxyl group of oxime, diaziridine formation with liquid ammonia, and a final step of oxidation with activated MnO_2_ to obtain diazirine. The chiral center configuration was checked for every step, and asymmetric 2-, 3-, and 4- trifluoromethyldiazirine-based lactisole **31** were able to be synthesized with fine yields.

#### 3.1.2. Diaziridine as an Intermediate for “Minimalist” into “All-in-One” 3*H*-Diazirine

The term “minimalist” aliphatic diazirine-based probe for photoaffinity labeling was popularized by Yao’s group in 2013 [47], and while it was describes as “minimalist” diazirine linkers in a recent review [48]. The first generation [47] of so-called “minimalist” diazirine-based linkers was able to minimize the interference towards target binding and was built upon an alkyl diazirine and a terminal alkyne connected by short aliphatic chains [47,49,50]. These “minimalist” linkers **34** were made possible by synthesized representative aliphatic ketone containing alkyne **35**, which had previously been transformed from ethyl acetoacetate **36** as the starting material (Scheme 4A). The minimum length was limited to 2 carbons on both sides of the diazirine moiety in the linker in order to enable the linker to possess different functional groups and to prevent any side reaction during the requisite NH_3_/NH_2_SO_3_H step in diaziridine **37** formation. Subsequently, the first-generation “minimalist” diazirine linker’s **34** alkyne terminal was changed into cyclopropenes, which are similar in size, by rhodium-catalyzed reaction of α-diazo esters with alkynes, which have since been claimed to be the second generation of minimalist linkers [51,52].

The strategy for the adjustment of fluorine into the “minimalist” aliphatic diazirine linker can be initiated by the reaction of ester **38** and monobromodifluoroalkyne **39** in order to form an equimolar mixture of difluoroketone **40** along with in situ formation of its hydrate **41**, depending on the electrophilicity of the difluoroketone (Scheme 4B). This 1:1 mixture is then treated with hydroxylamine, followed by tosyl chloride in pyridine and ammonia in sequence [53], thus forming difluorodiaziridine **42** as an intermediate prior to oxidation with the I_2_-Et_3_N system, which results in 3*H*-diazirine **43**. From this approaches, the trend of “minimalist” diazirine-based compounds has shifted towards: (1) the synthesis of aliphatic 3*H*-diaziridine via activation of tosyl oxime formation, in contrast to the conventional methods; and (2) addition of difluoro moiety into the “minimalist” diazirine-based compounds as an approach to mimicking the superior stability and selectivity [54] of 3-phenyl-3-(trifluoromethyl)-3*H*-diazirine (TPD).

The compact, “all-in-one” structure of the diaziridine-mediated radioisotope-free photoaffinity labeling probe can be categorized as a “minimalist” aliphatic 3*H*-diazirine that includes the aromatics, as well as the addition of the fluorine moiety. The parts of the “all-in-one” photoaffinity labeling probe comprise a diazirine as a photo-crosslinker (a carbene-generating group), a fluorinated carbon at the aliphatic 3*H*-diazirine side chain of the aromatics for mimicking TPD, and an alkyne [55] or azide [56] as a tag for the attachment of a detectable group via bioorthogonal click chemistry. “All-in-one” 3*H*-diazirine **44** containing both alkyne and azide are constructed from representative ketone **45** containing aromatics, fluorine moiety, and alkyne or azide terminal in advance, before transformation into the diaziridine **46** intermediate via tosyl oxime formation. The Swern oxidation—dimethylsulfoxide and oxalyl chloride followed by the addition of triethylamine—can be used for the oxidation of the diaziridine **46** intermediate instead of Ag_2_O-Et_2_O, resulting in the “all-in-one” 3*H*-diazirine **44** (Scheme 4C).

**Table 2 molecules-26-04496-t002:**
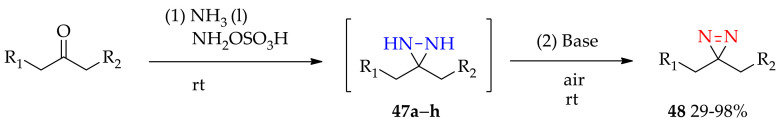
Diaziridine in-situ formation in one-pot aliphatic 3*H-*diazirine 48 syntheses.

Entries	Possible Diaziridine Formation (In-Situ)	Base (equiv.)	Reference
1	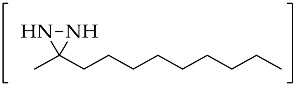	*t*BuOK (2.3 equiv.)	[57]
**47a**	KOH (2.3 equiv.)	[58]
2	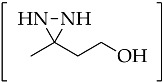	*t*BuOK (2.3 equiv.)	[57]
**47b**	KOH (2.3 equiv.)	[58]
3	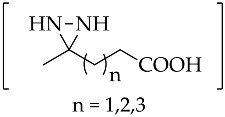	*t*BuOK (3.3 equiv)	[57]
**47c**	KOH (3.3 equiv.)	[58]
4	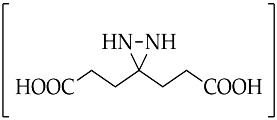	*t*BuOK (4.3 equiv.)	[57]
**47d**	KOH (4.3 equiv.)	[58]
5	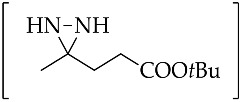	KOH (2.3 equiv.)	[58]
**47e**
6	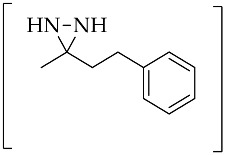	*t*BuOK (2.3 equiv.)	[57]
**47f**	KOH (2.3 equiv.)	[58]
7	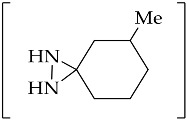	*t*BuOK (2.3 equiv.)	[57]
**47g**	KOH (2.3 equiv.)	[58]
8	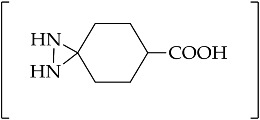	*t*BuOK (3.3 equiv.)	[57]
**47h**	KOH (3.3 equiv.)	[58]

#### 3.1.3. In Situ Formation of Diaziridines in One-Pot Synthesis of 3*H*-Diazirine

The dehydrogenation of the NH–NH bond in aliphatic diaziridine is a crucial transformation for generating the N=N double bond that acts as the main skeleton of aliphatic 3*H*-diazirine [57]. In the conventional method, different oxidants, including silver oxide or iodine in the presence of triethylamine, are introduced in order to transform diaziridines into 3*H*-diazirine following the removal of the ammonia [14,58,59,60]. Diaziridines are basic and form salts, and in several cases, the pure form of diaziridines can be difficult to isolate [59]. During the synthesis of 3*H*-diazirine as a photoaffinity labeled-based photophore, 3*H*-diazirine can be readily used for the further post-functional synthesis of the corresponding probe [8], and “bypassing” the prior synthesis steps of 3*H*-diazirine is the ideal way to run. Thus, the strategy for the current development of 3*H*-diazirine synthesis is the in situ formation of diaziridine, and approaches for one-pot synthesis of 3*H*-diazirine take advantage of this more comprehensive method.

Aliphatic diaziridine is generally synthesized from ketone treated with liquid ammonia to form the imine, followed by reaction with hydroxylamine-*O*-sulfonic acid (HOSA, NH_2_OSO_3_H), enabling intramolecular cyclization to the diaziridine. The addition of a base to the synthesis process of conventional aliphatic diaziridine from aliphatic ketone can provide the direct dehydrogenation of the in situ formed diaziridine into 3*H*-diazirine. The alternative method for dehydrogenation of hydroazobenzene, combining the least expensive bulk chemicals of liquid NH_3_ as a solvent and the readily available *t*BuOK system as a base, could be applied for the dehydrogenation of diaziridine [57]. Following this improvement, the use of base that can be easily handled and stored, as well as being highly available, such as KOH could enable gram-scale production [58]. Table 2 (Entries 1–8) shows the representative in situ formation of diaziridine **47a**–**h** for the synthesis of 3*H*-diazirine **48**. Kinetic study of the KOH-mediated direct transformations of diaziridine **47f** into 3*H*-diazirine **48** (Table 2, Entry 6) based on ^1^H-NMR analysis indicated that diaziridine possessed appropriate reactivity for ready transformation into 3*H*-diazirine [58]. When adding HOSA to liquid ammonia in ketone at room temperature for 12 h, the corresponding diaziridine **47f** (Table 2, Entry 6) was detected, as well as a trace amount of its 3*H*-diazirine form. Upon further treatment with KOH for 2 h, the 3*H*-diazirine was dramatically improved, indicating that diaziridine, as a precursor, can be converted directly, with the detection of no other by-products.

A carbene precursor, 3-phenyl-3-(trifluoromethyl)-3*H*-diazirine (TPD) derivatives are known for their ability to not promote cell death in the generation of active species in in vitro photoaffinity experiments [8]. Unlike the conventional synthesis of 3*H*-diazirine from diaziridine as an intermediate, the multistep reaction of TPD derivatives is performed from a phenyl trifluoromethyl ketone, which is transformed into the active species tosyl oxime, which is treated with an ammonia-Et_2_O solution at −78 °C, then brought to room temperature to allow the formation of diaziridine, which is oxidized with the I_2_-Et_3_N system as a final step to result in TPD [7,14,54]. Since the treatment of TPD after oxidation is time-consuming and yield-diminishing, the inevitable isolation of diaziridines can be skipped by using an alternative one-pot synthesis starting from tosyl oxime [61]. 

The treatment of tosyl oxime **49** with conventional reagents, depending on the temperature and the reaction time, can result in the formation of diaziridine **50** and/or its direct transformation into 3*H*-diazirine **51** [61]. The proposed mechanism for the transformation of tosyl oxime into diaziridine in past decades described the formation of the intermediate *gem*-diamine [62] following attack by NH_3_, and by removing the tosyl group and the proton, diaziridine is formed [61]. The current developments suggest that the neceesity of using liquid ammonia for the generation of NH_2_^−^ species can elongate the deprotonation of diaziridine to form diazirine. The self-ionization of liquid NH_3_ into NH_2_^−^ species seems crucial for this reaction, as demonstrated by the search for additives to increase this species, which has determined in studies that LiNH_2_ is superior to the inhibitor of NH_4_Cl (Figure 6). It should be noted that diaziridine formation is crucial, and the aim to achieve diaziridine isolation by removing the ammonia from the reaction, in terms of the conventional method, overlooks the important involvement of ammonia in the transformation of diaziridine **50** into 3*H*-diazirine **51** using tosyl oxime **49** as the starting material.

In order to broaden the applications of diaziridine-intermediated **52**, and the one-pot synthesis TPD **54** from tosyl oxime **53**, there are several viable substituents in the aromatic moiety of the starting material. Optically pure (trifluoromethyl)diazirinylphenylalanine ((Tmd)Phe) has been used as important building block in synthesis involving tosyl oxime as a precursor following the one-pot reaction method [61]. Method A prolongs the optical retention of (Tmd)Phe **55** and **56**, while the addition of LiNH_2_ in Method B results in the racemization of the product **55**. (Tmd)Phe **57**, deuterated at the aromatic moiety, was synthesized via Method A, and its optical activity was retarded. Furthermore, the synthesis of 3-nitro **58** or 3-aminol **59** and 3-/4-methyl TPD **60**–**61** for the construction of phenylthiourea derivatives [63] and saccharin derivatives [64], respectively, using a one-pot (method A or B) system resulted in an almost quantitative yield compared with the conventional stepwise Method C (Figure 7).

#### 3.1.4. Improved Method for the Synthesis of 3-[3-(Trifluoromethyl)-3*H*-diazirin-3-yl]aniline Derivatives

The needs of the aniline skeleton as a precursor for the multistep synthesis of the 3-phenyl-3-(trifluoromethyl)-3*H*-diazirine-based photoaffinity labeling probe faces an obstacle with respect to obtaining high overall yields. Studies describing the synthesis methods employed in previous decades [65,66,67,68] provide limited experimental details or analytical data; thus, it has become necessary to improve on these methods. In a brief (Figure 8), recent study has successfully synthesized 3-[3-(trifluoromethyl)-3*H*-diazirin-3-yl]aniline **62** with excellent yield through the reduction of nitrobenzene derivatives following the formation of 3*H*-diazirine **63a**–**b** [63] or the protection of the amino group from undergoing 3*H*-diazirine formation, followed by its deprotection, resulting in **63c**–**e** [63,69]. This new improvement showed that it is preferable to protect the amino group at the 3-position with —Boc (**63c**), enabling convenent acidic deprotection after 3*H*-diazirine formation [63,69]. Meanwhile, the amino group at the 4-position can be subjected to two types of protecting group (—Boc **63d** or —phthalimide **63e**), and in the case of —phthalimide group (**63e**), this protecting group can be directly oxidized in conjunction with the diaziridine **64** moiety by MnO_2_-CH_2_Cl_2_ into a final product of 3-[3-(trifluoromethyl)-3*H*-diazirin-3-yl]aniline derivative **62** [63]. 

A few studies have performed the final step of forming the 3-[3-(trifluoromethyl)-3*H*-diazirin-3-yl] aniline derivatives from nitrobenzene derivatives through the reduction of the nitro moiety in sodium dithionite [68,70]. During the synthesis of photoreactive 2-propoxyaniline derivatives as artificial sweeteners [71], the reduction by sodium dithionite after 3*H*-diazirine formation did not take place, and the diazirinyl moiety was unable to tolerate this condition, causing decomposition to occur. A possible route for this was diaziridine **65** formation following reduction with sodium dithionite (Figure 8). A moderate yield was obtained without breaking the diaziridinyl moiety, and validated the essential aspects of diaziridine formation, enabling a better understanding of the improved method for NH_2_-substituted 3-phenyl-3-(trifluoromethyl)-3*H*-diazirine **62**.

#### 3.1.5. Expansion of Fluorous 3*H*-Diaziridine as a Basis for 3*H*-Diazirine Application

A new fluorous 3*H*-diazirine-based photoaffinity labeling probe taking advantage of fluorous chromatography and fluorous solid phase extraction was developed. These techniques aim to retard fluorinated compounds on fluorinated silica gel, while all other organic material is washed off the column with the appropriate organic mobile phase. A new fluorous 3*H*-diaziridine-based photoaffinity labeling probe, with a longer perfluoroalkyl residue substituted for the trifluoromethyl group in the non-radioactive photoaffinity labeling probe, can be synthesized starting from the transformation of perfluoroalkylated ketone **66** (perfluorobutyl and perfluorooctyl (Figure 9A) [72]) into diaziridine **67** through the conventional approach of oxime formation, tosylation, and treatment with liquid ammonia. The temporary silylation of the diaziridine moiety of compound 68 was performed by means of trimethylsilyl triflate, bromide–lithium exchange, and carboxylation with carbon dioxide for the conversion of benzoic acid. The final product was observed after the oxidation of diaziridine **68** into diazirine **69** with I_2−_Et_3_N. Prior to the study reporting the perfluoroalkylated 3*H*-diaziridine-based photoaffinity labeling probe, the perfluoropropyl and perfluorohexyl [73] residues had also been studied using the conventional approach, starting from the modification of perfluoroalkylated ketone **70** to result in diaziridine **71**, which can readily be oxidized to 3*H*-diazirine 72 (Figure 9B).

Diazirine-based crosslinkers (synthesized from the counterpart diaziridine) have been explored due to their ability to act as carbene-generating reagents. Since the 3-trifluoromethyl 3*H*-diazirine can be photoreactivated with light at 350 nm to generate carbene, this photophore is categorized as a high reactive species, and is able to rapidly form crosslinks to biomolecules with short photoirradiation times [8]. This universal crosslinker for aliphatic polymers, the simple bis-diazirine reagent, contains the known compound 1,3-bis(3-(trifluoromethyl)-3*H*-diazirin-3-yl)benzene or its pyridyl analog **73**, which is volatile, and is prone to explosion (Figure 10A) [74]. Thus, the bis-diazirine crosslinker was improved by the addition of an electron-deficient linker at the *para* position of diazirine compound **74** [74], which can be handled under ambient conditions, preventing any self-reaction due to the absence of any aliphatic C-H bonds (Figure 10B). The so-called second-generation bis-diazirine **75** is an elongated perfluoroalkylated chain inspired by the first-generation bis-diazirine **74** [75]. The highly fluorinated bis-diazirines designed in the study can be developed into a flexible covalent adhesive (Figure 10C). The simple 1st-generation and 2nd-generation bis-diazirines are highly useful as crosslinkers for a broad range of feasible synthesis applications applications starting from representative ketone **76**–**78** into diaziridine **79**–**81**, which can be directly oxidized by the I_2−_Et_3_N system.

#### 3.1.6. Unconventional Non-Substituted Diaziridine Synthesis Approaches

Recently, there have been two methods that have taken advantage of the conventional concept and turned it into brand new approaches to developing diaziridine synthesis. The first is the use of resin-bound sulfonyl oximes for the synthesis of 3-trifluoromethyl-3-phenyldiaziridine [76]. A commercially available polystyrene–sulfonyl chloride **82**, used as an equivalent to replace the common reagents mesyl or tosyl chloride, was previously reacted with oxime **83** to provide the desired solid-supported sulfonyl oxime. The conventional-like active species sulfonyl oxime **84**–**86** were then transformed by ammonia/dioxane solution to obtain a set of substituted diaziridines **87**. The presence of the trifluoromethyl group has a role in the cleavage of the immobilized sulfonyl oxime precursors from their solid supports, therefore allowing good results for diaziridine **87** formation. The compatibility of the immobilization and cleavage protocols in this method was tested with several aromatic building blocks and alkyne groups (Figure 11). An optical retention study of the optical active-aromatic substituents or building blocks remains to be performed for this method in the future.

The second unconventional method is the ammonia-free synthesis of 3-trifluoromethyl-3-phenyldiaziridine **88** [77]. The study started with the preparation of trifluoromethyl phenyl imines **89** using lithium bis(trimethylsilyl)amide [78] into trifluoromethyl phenyl ketone **90**, resulting in *N*-TMS-ketimine **91** (Figure 12). The solvolysis of this *N*-TMS-ketimine with methanol provides the representative imine for direct diaziridine formation. Since the use of the conventional reagent HOSA, which acts as the *O*-sulfonyl hydroxylamine source, did not yield the desired diaziridine product due to the reduced nucleophilicity of the amine function of the formd zwitterions, this study therefore then reacted *p*-toluenesulfonyl hydroxylamine with trifluoromethyl phenyl imines **89** as an advanced ammonia-free methodology for the synthesis of diaziridine **88** (Figure 12). The ammonia-free synthesis was carried out in significantly fewer reaction steps and with less purification of the intermediate compared to the multiple overnight steps of the conventional method. Various 3-trifluoromethyl-3-phenyldiaziridine derivatives can be explored for further study based on ammonia-free methods, despite the difficulty of synthesizing the imine intermediate. In addition, both of the unconventional methods described here successfully oxidized the resulting 3-trifluoromethyl-3-phenyldiaziridine into representative diazirine using the common oxidation reagents MnO_2_ [76] or Ag_2_O [77].

### 3.2. N-Monosubstituted Diaziridines

#### 3.2.1. Conventional Method for Recent Uses

In general, conventional methods for the synthesis of diaziridines with HOSA have been performed in various solvents, such as water [79], methanol [80], or pure liquid ammonia [81]. When conventional methods were used for the reaction of cyclooctanone **92**, benzylamine **93**, and HOSA, the corresponding bicyclic diaziridine, 1,2 diazaspiro[2,7]decane **94** (Figure 13), was produced with quite a low yield (~20%) [81]. In order to achieve a better understanding of this, the solvent dependency of the *N*-monosubstituted diaziridine **94** yield was studied by means of GC–MS [82]. The highest yield of the product *N*-monosubstituted diaziridine **94** (~42%) corresponded to the use of an apolar aprotic solvent, such as cyclohexane or toluene.

The reactions of nucleophilic diaziridines with electrophilic alkynes were established decades ago [83], and by means of the ^15^N-labeling experiment, it was shown that the mechanism of the reaction could be assumed to involve the initial addition of an alkylated nitrogen atom (*N*-monosubstituted diaziridine counterpart) to provide an intermediate that could subsequently undergo ring-opening and proton transfer, resulting in the adduct product [84]. The trapping reaction resulted in thermally generated benzyne species **95a**–**f** with heteroatom-rich diaziridine **96** leading to *N*-arylated hydrazones **97a**–**f** in a single step, while these could be converted into fused-ring indole derivatives **97c**–**e** in some cases (Scheme 5). The conventional method for the synthesis of diaziridines has been used in recent years to form *N*-ethyl diaziridine **98a** in quantitive yield from the reaction of 2-adamantane-2,3-[3*H*]-diazirine **99** with alkyl Grignard reagents (Scheme 6). Despite this success, the use of aryl Grignard reagents did not lead to N-phenyl diaziridine, and the expected diaziridine **98b** was obtained only upon the addition of phenyl lithium. The direct reaction with acetylacetone was then conducted from the pure resulting *N*-phenyl diaziridine **98****b**, allowing pyrazole **100** and adamantanone **101** to be obtained with excellent yields (Scheme 6) [85].

#### 3.2.2. Base Addition for Enhancement of N-Monosubstituted Diaziridine Formation

It is known that the formation of monocyclic diaziridine the three-component condensation of carbonyl compounds, primary aliphatic amines, and HOSA in protic medium is substantially dependent on pH value. A possible pathway for this is through the generation of carbenium-iminium cations, which subsequently react with the aminating reagent (HOSA) to form an intermediate compound of an aminal type. The presence of base would rapidly cyclize this intermediate to result in diaziridine [20]. The addition of base to the process was followed by diastereoselective synthesis of *N*-monosubstituted diaziridines coupled with neurotransmitter amino acid **102** [86]. The pH can be adjusted with the simultaneous addition of HOSA (1 equiv.) and 30% aqueous NaOH in order to neutralize the H_2_SO_4_ that formed during the reaction, and the requirement vary depending on the substrate. Moreover, the new hybrid structure of *N*-monosubstituted diaziridine **102** can be prepared from the reaction of aldehydes or ketones **103** and amino acid ethyl esters **104** under mild conditions, with the major diastereomer corresponding to the racemic mixture of two meso-forms, 1*R**, 2*R**, 3*S** (Scheme 7).

The use of *N*-monosubstituted diaziridines as *N*-transfer reagents to α,β-unsaturated amides to form stereopure aziridines has been reported [16]. Taking advantage of the reactivity of diaziridine as an essential precursor, diastereoselective synthesis of *N*-monosubstituted diaziridine **105** can be conducted from simple aldehydes or ketones **106** with amines **107** [87]. The reaction takes place in the presence of hydroxylamine *O*-sulfonic acid (HOSA), a conventional aminating reagent. The addition of NaHCO_3_ results in a major product of *N*-monosubstituted diaziridine **105**, suppressing the formation of *N*-monosubstituted imine (**108**, Figure 14). This weak inorganic base also showed high product yield compared with K_3_PO_4_ or Et_3_N, and can be used to replace an excess of amine counterparts. The method provides the resulting *N*-monosubstituted diaziridines **105** with a single diastereomer and a wide variety of aromatic and aliphatic aldehydes or ketones **106** and amines **107**.

#### 3.2.3. Unconventional N-Monosubstituted Diaziridine Synthesis Approach

When an excess of the mixture of CS_2_ and KOH was added to hydrazide **109** spirit boiling solution, the hydrogen sulphide evolved in the reaction. As a result of the elimination of hydrogen sulphide, three-membered diaziridine **110** was formed, along with another product, 1,3,4-thiadiazolidine [2]. Although the amounts of CS_2_ and KOH were varied, the same substances were formed, with the amount of CS_2_ affecting only the yield of the final product. The maximum reaction yield was observed when hydrazide, CS_2_, and KOH were taken in a molar ratio of 1:1.7:2.0, indicating that diaziridine **110** had been formed, while for thiadiazolidine cycle formation, an excess CS_2_ of more than double is required. At first, it is possible to form potassium salt, which is then converted into diaziridine derivative under the action of hydrochloric acid. Using this unconventional method, compound **110** can be either thion or thiol tautomeric (Figure 15A). Using the described method, the 1-(4-methoxy-6-methyl- pyrimidin-2-yl)-diaziridine-3-thion **111** can also be established from hydrazide **112** (Figure 15B). As for the influence of the carbonyl group on the reaction, similar interactions were carried out using 4,6-bis-dimethylamino- [1,3,5]triazine-2-carboxylic acid hydrazide **113** (Figure 15C) and 3,4-di- methyl-2-thioxo-2,3-dihydro-thiazole-5-carboxylic acid hydrazide **114** (Figure 15D). The *N*-monosubstituted diaziridine of (4,6-bis-dimethylamino-[1,3,5]tria- zine-2-yl)-(3-thioxo-diaziridin-1-yl)-methanone **115** and (3,4- dimethyl-2-thioxo-2,3-dihydro-thiazole-5-yl)-(3-thioxo- diaziridin-1-yl)-methanone **116** can be obtained (Figure 15C,D).

### 3.3. N,N-Disubstituted Diaziridines

#### 3.3.1. Conventional Method for Recent Uses

*N*,*N*-disubstituted diaziridines containing trifluoromethyl at the 3-position **117** can be synthesized using conventional conditions. Various trifluoromethyl imines **118** have been used in amination reactions with NsONHCO_2_Et as the aminating agent [88]. A twofold excess of NsONHCO_2_Et in CH_2_C1_2_ without any addition of base at room temperature can give the corresponding *N,N*-disubstituted diaziridines **117** (Scheme 8A). *N*,*N*-Disubstituted diaziridines **119** were also successfully synthesized by the reaction of trifluoromethyl imines **120** with NsONHCO_2_*t-*Bu, a carbamate, which is known to give amination reactions only by an ionic pathway involving the corresponding aza-anion (Scheme 8B). The advantage of using conventional methods for the synthesis of *N*,*N*-disubstituted diaziridines was also observed when synthesizing bidiaziridine **121** from α-diimines **122** [89]. A two-phase H_2_O/CH_2_Cl_2_ system and the portion-wise addition of both amination agent and base (NsONHCO_2_Et and CaO) is considered to favor the formation of the monodiaziridine (E)-3-(iminomethyl)diaziridine-1-carboxylates **123**, rather than its bidiaziridine **121** (Figure 16A). Further functionalization of this monodiaziridine, **123a,c,d,** can be performed to obtain a hybrid compound containing an diaziridine and oxaziridine ring in one molecule of 3-(diaziridin-3-yl)oxaziridines **124** (Figure 16B) [89].

In line with the synthesis of the hybrid compound, a diastereoselective method for the synthesis of diaziridines with *N*- and/or *C*- cyclopropyl substituents (**125** and/or **126**) was developed [90] using the conventional methods. A one-pot, three-component condensation of cyclopropyl-containing carbonyl compounds **127**, primary aliphatic amines, including cyclo propylamine **128**, and *N*-chloroalkylamines **129** was conducted in the organic solvents. This typical reaction was carried out under mild conditions and in the presence of bases (Scheme 9), resulting in the predominance of diaziridine product **125** and/or **126**, with the major diastereomer being the racemic mixture of two meso-forms, and the racemic mixtures of two enantiomers for the minor diastereomer. The use of K_2_CO_3_ as a base, and an aprotic solvent (CHCl_3_) obtained from equimolar amounts of aldehyde, cyclopropylamine, and pre-synthesized *N-*chloroalkylamines can provide access to compounds with cyclopropyl substituents on the nitrogen atom of the diaziridine ring. As for the enantio- and diastereoselective synthesis of diaziridines **130** via aziridination of *N*-tosyl aldimines **131** while applying modified hydroxylamine **132** under asymmetric phase-transfer catalysis **133**, it was recognized that K_2_CO_3_ was too weak to mediate this reaction [91]. Thus, K_2_PO_4_ was then utilized, and the reaction rate was significantly enhanced, and gave diaziridine with similar selectivity. By taking advantage of a phase-transfer catalyzed nitrogen insertion into the π-system of *N*-tosyl aldimines **131** and the use of *N*-transfer agent of *N*-benzyl-*O*-benzoyl hydroxylamine **133**, the afforded *N*-tosyl-*N*′-benzyl-diaziridines **130** can be synthesized as a single diastereomer in up to high yields with high to excellent enantioselectivity (up to 96% ee, Scheme 10).

#### 3.3.2. The Green Transformation of 6-Aryl-1,5-diazabicyclo[3.1.0]hexanes

Green chemistry is leading the development of ionic liquids as alternative solvents that possess useful physicochemical properties, such as non-inflammability, low vapor pressure, and ease of regeneration. These trends have led to the transformation of *N*,*N*-disubstituted diaziridines under ionic liquid solvent-based reaction. The 6-aryl-1,5-diazabicyclo[3.1.0]hexanes **134** can be thermally (by refluxing in toluene or xylene) or catalytically (by the addition of 20 mol % Lewis acid (BF_3_∙Et_2_O)) transformed into active species of azomethine imines **135** (Figure 17). In [92,93], mild conditions were represented by the use of ionic liquid medium, and it was found that the reaction could be managed by ionic liquid, rather than the organic solvent, of MeCN [94]. In some case, heating was required (40 °C or 50 °C) to to initiate the reaction with olefins **136**, which resulted in the [3 + 2] cycloaddition products of 1,5diazabicyclo[3.3.0]octane derivatives **137**. High diastereoselectivity can be achieved through the [3 + 2] cycloaddition of azomethine imines derived from 6-aryl-1,5-diazabicyclo[3.1.0]hexanes to acrylonitrile and 4-nitrophenyl vinyl sulfone in ionic liquid of [emim][OTf] [93]. The transformation of diaziridine **134** into active species of azomethine imines 135 showed different reactivity when reacted with arylidenemalononitrile **138** (Figure 17). The presence of an electron-withdrawing NO_2_ substituent in the aromatic ring of arylidenemalononitrile, as the starting material, resulted in pyrazoline **139**, and the isolation of arylidenemalononitriles containing aromatic fragments of the starting bicyclic diaziridines **140** was also achieved [95]. The annulations of azomethine imines **135**, derived from the representative diaziridine **134** with 1*H*-indole-2,3-diones **141**, can occur on the basis of the reaction between azomethine imines and carbonyl compounds. There are two types of product introduced in the ionic liquid of [bmim][BF4] systems: pyrazolines **142**, which are proposed to be proced by the induction of a formal 1,4-H shift in azomethine imine; and pyrazoles **143** (Figure 17), which are most likely to emerge as a result of the oxidation of pyrazolines by air oxygen [96]. Moreover, the reaction between 6-aryl-1,5-diazabicyclo[3.1.0]hexanes and 1,3-diarylpropenones was also conducted under microwave irradiation (110 °C), which is compliant with the principles of green chemistry in organic synthesis [97]. The [3 + 2] cycloaddition products of perhydropyrazolopyrazoles were formed in fine yields as single diastereomers.

#### 3.3.3. Metal Catalysis of Diaziridines Ring Opening

A single reagent that is able catalyze multiple mechanistically distinct processes in a chemical reaction known as an auto-tandem catalyst [98], and the transition metal used for this process is gold(I). The ring opening of diaziridines [99] is started by the synthesis of *N*-monosubstituted diaziridine by the conventional method of reacting ketone **144**, benzyl, and the aminating reagent NH_2_OSO_3_H. A minimal modification at the free *N*-terminal of diaziridine resulted in 1-benzyl-1,2-diazaspiro[2,5]octane **145**, which was then used as the starting material for an auto-tandem catalysis reaction mediated by gold(I). The model reaction was conducted by reacting diaziridine **145** with phenylacetylene **146**, and it was found that the product 3-pyrazoline **147** was preferentially obtained using Ph_3_AuNTf_2_ as the catalyst, rather than other catalysts. The alkyne insertion product **146** was possibly an intermediate in the process, since it can be directly transformed into the product 3-pyrazoline **147** by the use of Ph_3_AuNTf_2_. Taken together, gold(I) as an auto-tandem catalyst is capable of being used to pursue the typical subsequent ring opening of diaziridine **145** and the cyclization of **148** (Figure 18).

The [3+3]-annulation of bicyclic diaziridines **149** with *N*-substituted aziridines **150a,b** in iron salt can be processed into [1,2,4]-triazines **151a,b** as a single diastereoisomer by using FeCl_3_ (Scheme 11). The chelation of aziridines with FeCl_3_ can lead to stereospecific ring opening using azomethine imine intermediate derived from diaziridine and the diasteroselectivity potentially obtained by the steric factors during the approaches of aziridine with the azomethine imine. Diaziridine **149** was tested using optically active aziridine **150b**, demonstrating the stereoselectivity of this typical iron-catalyzed reaction [100]. The one-pot reaction of bicyclic diaziridines **152** and (2-bromo-2-nitrovinyl)arenes **153** resulting in bicyclic cationic **154** (2,3-dihydro-1H-pyrazolo[1,2-a]pyrazol-4-ium as cation and [Ce^3+^(NO_3_)_6_]^3−^ as anion) can only be conducted under the action of ceric ammonium nitrate (CAN) and MeCN (Figure 19). CAN as the catalyst can trigger in situ diaziridine ring opening of bicyclic diaziridines or metathesis of azomethine imine, the occurrence of [3+2] cycloaddition of (2-bromo-2-nitrovinyl)arenes, followed by the aromatization of the formed pyrazolidine ring [101]. This metal catalysis of diaziridine ring opening approaches (Scheme 11 and Figure 19) provides the advantage of simplicity, and is a powerful tool for the synthesis of hydrazine bicyclic compounds.

#### 3.3.4. Diaziridine Reaction with Donor–Acceptor Cyclopropanes/Cyclopropenes

The first [3+3]-annulation of two different three-membered rings of bicyclic diaziridines **154a**–**n** and donor–acceptor cyclopropanes **155** was reported in 2018 [102]. The reaction afforded perhydropyridazine derivatives **156** in high yields and diastereoselectivity under mild Lewis acid catalysis of Ni(ClO_4_)_2_·6H_2_O (Figure 20). Under the same conditions, the difference of alkyl substituent(s) at the C(6) atom of bicyclic diaziridine **154k,o**–**u** can prevent (3+3)-annulation. The spiro-[cyclohexane-1,6′-(1,5-diazabicyclo[3.1.0]hexane)] and other bulky substituents at the C(3) atom of the diaziridine ring of 1,5-diazabicyclo[3.1.0]hexanes (**154k,o**–**u,v**) did not produce the corresponding hexahydropyridazine **156** and afford 1-alkylated 2,3-dihydropyrazoles **157** or **158** in the reaction with donor–acceptor cyclopropanes **155** [103]. The reaction possibly proceeds via the alkylation of a diaziridine derivative with a Lewis acid-activated cyclopropane, followed by hydration of the formed 1,6-zwitterion, producing hemiaminal. Following this process, the elimination of the carbonyl compound and the air oxidation of the formed pyrazolidine accomplishes the synthesis of 1-alkylated 2,3-dihydropyrazole derivatives (**157** or **158**, Figure 20).

Since diaziridine is a strained ring system that can undergo N–N bond cleavage, the ring opening of bicyclic diaziridine **159** was continued for its reaction with enoldiazoacetates **160**, which resulted in the cycloadduct **161** (Scheme 12). The metallo-enolcarbenes was initially generated to undergo the novel [3 + 2]-cycloaddition. The asymmetric synthesis process is able to produce the cycloaddition product **161** with 31% ee or 95% ee, depending on the isomeric mixture of the starting material. The metallocarbene source of donor–acceptor cyclopropene **162** is also able to form the cycloaddition product **161** in 95% ee (Scheme 12), suggesting that the initially generated cycloaddition of metallo-enolcarbenes is possible at a faster rate than the formation of the corresponding donor–acceptor cyclopropene **162** [104].

#### 3.3.5. Unconventional *N*,*N*-Disubstituted Diaziridine Synthesis Approach

An unconventional nonmetallic and photocatalytic approach to *N*,*N*-disubstituted diaziridine synthesis was developed [105], excluding hydroxylamine-*O*-sulfonic acid [20] and ethyl nosyloxycarbamate [18] for the mediation of condensations of carbonyl compounds, amines, or ammonia. A blue LED, an organic photocatalyst (rose Bengal **163**), and a Lewis acid such as the oxidant PhI(OAc)_2_ **164** were used to generate functionalized diaziridines with excellent reaction rates and yields that also showed high stereoselectivities for the synthesis of diaziridines **165** with chiral substituents. The reaction proceeded through the in situ formation of nitrene **166** and an imine **167** intermediate, whereby their formation involved the light-activated Rose Bengal in order to convert related species from the combination of PhI(OAc)_2_
**164**, amine **168**, with 1,2-diol **169** or the simultaneous combination of PhI(OAc)_2_ directly with amine. The typical *N*,*N*-disubstituted diaziridine **165** synthesized using this novel and extraordinary method is summarized in Scheme 13.

## 4. Conclusions

The non-, *N*-mono-, and *N*,*N*-disubstituted diaziridines are attractive skeletons as building blocks in the synthesis of important intermediates and precursors with excellent stereochemical properties. The trend of diaziridine formation is still following the conventional concept, wherein valuable synthesized diaziridine derivatives are added together with various substituents. More unconventional approaches are limited to the period of the last decade, and further studies aiming to extend these unconventional methods remain to be performed. The study of diaziridine transformation in the field of organic and pharmaceutical chemistry—especially to become a precursor in diazirine-based photoaffinity labeling probes for drug design—as well as its stereochemistry, offer great opportunities for further exploration.

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
