# Peer review of "New Trends in Diaziridine Formation and Transformation (a Review)"

_molecules, 2021, doi:10.3390/molecules26154496_

Round 1

Reviewer 1 Report

In general, the manuscript was well discussed.

1) In "...three research groups between 1958—1959,..." please give credit to groups (by naming them);

2) In Fig 4, reagents: "...Et2N" ... ???;

3) In the table 1: change "Conformations (Configurations)" to "Conformations/Configurations"

4) Check the number of Figures in the text, for example: Fig 3A, 3B (lines 173 and 178); line 228: Fig 4 (A), switch to Fig. 9, and so on;

5) Another mistake is in line 182:  "(Compound 47g−h, Tabel 1, Entries 7-8)",  it is about Table 2; 

6) In this article, the words diaziridine (259 times) and diazirine (141 times) are found. I think diazirine needs to be mentioned in the abstract, keywords, and in the conclusion. 

7) All figures that have reagents, conditions and yield need change to Schemes. Exemples: Fig 8, Fig 9, and Fig 11, etc. 

Author Response

We sincerely thank to the reviewer 1 comments and suggestions. We were pleased to know that our manuscript was rated as potentially acceptable for publication.

Reviewer 2 Report

The review manuscript by Tachrin and co-workers deals with the class of diaziridines concerning their synthesis and transformations. The work is well written and brings severalinterestingexamples about their preparation under unconventional approaches. Given the importance of this class of compounds, synthetic viability and good reactivity, the work can be valuable guide for the synthetic chemistry community. For this reason, I recommend the work for publication after minor English grammar checking.

Author Response

We sincerely thank to the reviewer 2 comments and suggestions. We were pleased to know that our manuscript was rated as potentially acceptable for publication.

Reviewer 3 Report

The manuscript titled “New Trends in Diaziridines Formation and Transformation (A Review) (Zetryana Puteri Tachrim, Lei Wang, Yuta Murai, and Makoto Hashimoto)” is very interesting and attractive to many researchers. This manuscript focuses on diaziridine, a high strained three-membered heterocycle with two nitrogen atoms, formation, and transformation. The research problems are well explained. The manuscript is clear and well structured. I recommend this manuscript for publication.

Author Response

We sincerely thank to the reviewer 3 comments and suggestions. We were pleased to know that our manuscript was rated as potentially acceptable for publication.

Here the reply notes according to some specific issues from reviewer 3.

In general, the manuscript was well discussed.

1) In "...three research groups between 1958—1959,..." please give credit to groups (by naming them);

Addition into the sentence

Diaziridines, independently discovered by three research groups (Schmitz’s, Paulsen’s, and Abendroth & Henrich’s research group) between 1958—1959, taken the popularity in the chemistry progressively in the next two decades after the first publications [5].

2) In Fig 4, reagents: "...Et2N" ... ???;

Et2N Change to Et3N

3) In the table 1: change "Conformations (Configurations)" to "Conformations/Configurations"

Table 1 change to Conformations/Configurations

4) Check the number of Figures in the text, for example: Fig 3A, 3B (lines 173 and 178); line 228: Fig 4 (A), switch to Fig. 9, and so on;

The changes were yellow highlighted

5) Another mistake is in line 182: "(Compound 47g-h, Tabel 1, Entries 7-8)", it is about Table 2;

The changes were yellow highlighted

6) In this article, the words diaziridine (259 times) and diazirine (141 times) are found. I think diazirine needs to be mentioned in the abstract, keywords, and in the
conclusion.

The addition of diazirine includes in:

Abstract

This review focuses on diaziridine, a high strained three-membered heterocycles with two nitrogen atoms that role as one of an important precursor of diazirine photoaffinity probe, formation and transformation.

Keywords

Diaziridines, substituted diaziridines, diazirines, formation, synthesis, transformation, reactivity

Conclusion

The study of diaziridine transformation in the field of organic and pharmaceutical chemistry—especially to become a precursor in the diazirine-based photoaffinity labeling probe for drug design—as well as its stereochemistry is having high opportunities to be explored.

7) All figures that have reagents, conditions and yield need change to Schemes. Exemples: Fig 8, Fig 9, and Fig 11, etc.

The changes were yellow highlighted